# Going Beyond Neural Network Feature Similarity: The Network Feature Complexity and Its Interpretation Using Category Theory

**Yiting Chen, Zhanpeng Zhou, Junchi Yan**[*]
Department of Computer Science and Engineering, Shanghai Jiao Tong University
{sjtucyt,zzp1012,yanjunchi}@sjtu.edu.cn
https://github.com/Ytchen981/Iterative-Feature-Merging

## Abstract

The behavior of neural networks remains opaque, and a recently widely noted phenomenon is that networks often achieve similar performance when initialized with different random parameters. This phenomenon has attracted significant attention in measuring the similarity between features learned by distinct networks. However, feature similarity could be vague in describing the same feature while similar features could yield different outputs. In this paper, we expand the concept of equivalent feature and provide the definition of what we call *functionally equivalent features*. These features produce equivalent output under certain transformations. Using this definition, we aim to derive a more intrinsic metric for the layer-wise *feature complexity* regarding the redundancy of features learned by a neural network at each layer. We offer a formal interpretation of our approach through the lens of category theory, a well-developed area in mathematics. To quantify the feature complexity, we further propose an efficient algorithm named Iterative Feature Merging (IFM). Our experimental results validate our ideas and theories from various perspectives. We empirically show that the functionally equivalence widely exists among different features learned by the same network and we could reduce network size almost without affecting the performance.

## 1 Introduction

Deep neural networks (DNNs) have achieved significant success across diverse fields, including vision, texts and other areas. However, DNNs are often regarded as "black box" models associated with high dimensional feature maps and numerous parameters. Recently, many studies report an interesting phenomenon that neural networks with different random initialization often converge to solutions with similar performance on the test set (Dauphin et al., 2014; Frankle et al., 2020). Meanwhile, methods such as pruning (Wen et al., 2016; Ye et al., 2018; Peng et al., 2019) and knowledge distillation (Hinton et al., 2015) have been proposed to reduce the complexity of the neural network structure, leading to a more compact DNN achieving similar performance to their dense/large counterparts. These phenomena further draw increasing studies on the features[1] learned by neural networks and a popular treatment is measuring the so-called feature similarity[2]. Various feature similarity measures (Li et al., 2016; Morcos et al., 2018) have been devised to quantify the distance between two features, and some works have further empirically found that similar features can be learned from either between different networks (Li et al., 2016; Wang et al., 2018) or within a (wide) network (Nguyen et al., 2021). In literature, feature similarity measures are often designed to be invariant to certain transformations under certain transformations *e.g.* permutation (Li et al., 2016), isotropic scaling (Barannikov et al., 2022), invertible linear transformation (Wang et al.,

---

[*]Correspondence author. This work was in part supported by NSFC (92370201, 62222607) and SJTU Trans-med Awards Research (STAR) 20210106.

[1]More precisely speaking, the feature here is the layer-wise feature map corresponding to a data point. Note that most previous works, *e.g.* measuring feature similarity, are based on a set of features corresponding to a specific dataset, while our definition and method are data-agnostic and may be more fundamental.

[2]The feature similarity is also referred to as representational similarity.

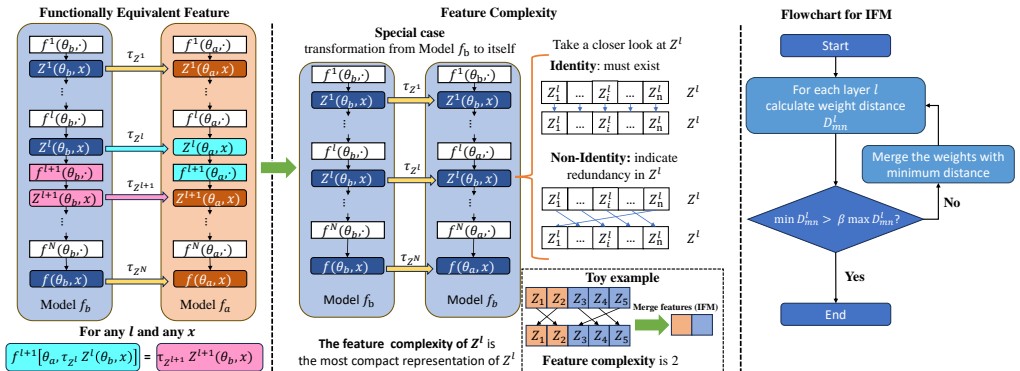

Figure 1: An illustration of our proposed functionally equivalent features and feature complexity. Two different models learn functionally equivalent features if the features and outputs of the two models are equivalent under certain invertible linear transformations. Feature complexity (layer-wise) is the dimensionality of the most compact representation of the feature at a certain layer. In order to retrieve the most compact version, we propose iterative feature merging (IFM).

2018), affine transformation (Raghu et al., 2017; Williams et al., 2021) etc. However, previous work (Ding et al., 2021) has shown similar features could also yield different outputs. It shows that previous feature similarity measures are often vague in exactly describing the inherent structure of the neural networks (*e.g.* the redundancy of features learned by a neural network or the same feature learned by different neural networks) and can hardly provide actionable insights.

In this paper, we expand the concept of feature equivalence and formally define a more general form, the so-called functional equivalence. Features are considered functionally equivalent if they produce outputs that are equivalent under certain transformations. Based on the functionally equivalent features, one can derive a metric to reflect the inherent layer-wise feature complexity of the network regarding the redundancy of features learned by neural networks. In another word, the feature complexity corresponds to the most compact representation of the feature at a certain layer.

For a clearer and more formal description of our approach, we provide an abstract view of neural networks and interpret our approach using the language of category theory, a branch of mathematics, which is widely applied in various fields, including mathematics, physics, and computer science. In general, a category is a graph with objects and arrows (or "morphisms") between objects. The mappings between categories are called functors that preserves the structure and the mappings between functors are called natural transformations (for a formal introduction, see Sec. 2.2). In the context of neural networks, we abstractly represent the network structure as a category and a certain neural network as a functor that maps this structure to specific parameters. Through the lens of category theory, we prove that functional equivalence between features can be elegantly represented as the existence of a natural transformation between two functors defined above.

To empirically measure the feature complexity, we propose an algorithm called Iterative Feature Merging (IFM) to merge the weights corresponding to redundant features. By iteratively matching the weights corresponding to different features, we merge functionally equivalent features. Experimental results show that DNNs learns a lot of functionally equivalent features which could be merged or removed with little impact on the neural network performance. We also provide many valuable insights that may inspire future works. **Our major contributions are as follows:**

• We provide a definition for functionally equivalent features (in Sec. 2.3) and the layer-wise feature complexity (in Sec. 2.4). This differs largely from existing works dwelling on feature similarity.

• We provide a category theory perspective on neural network structure and training. From the category theory perspective, we further offer insights into phenomena like linear mode connectivity.

• We propose an algorithm called Iterative Feature Merging (IFM) (in Sec. 3) to measure the feature complexity. Experimental results show its efficiency and potential as an effective pruning method. Experiments with IFM also yield valuable insights.

## 2 DEFINITION OF NEURAL NETWORK FEATURE COMPLEXITY

In this section, we define the feature complexity of a neural network from a category theory perspective. We first introduce an abstract view of neural networks using the language of category theory (in Sec. 2.2) and further define functionally equivalent feature from the category theory view (in Sec. 2.3). Finally, we provide the definition of feature complexity in Sec. 2.4.

### 2.1 NOTATIONS

Consider an $L$-layer neural network $f(\theta; \cdot)$ where $\theta$ is the parameter. We use $f^l(\theta^l, \cdot)$ to denote the $l$-th layer. For the feature map $Z^l(\theta, \mathbf{x})$, we have $Z^l(\theta, \mathbf{x}) = f^l(\theta^l, Z^{l-1}(\theta, \mathbf{x}))$ and $Z^1(\theta, \mathbf{x}) = f^1(\theta^1, \mathbf{x})$. When $\mathbf{x}$ is not of concern, we abbreviate $Z^l(\theta)$ for $Z^l(\theta, \mathbf{x})$.

To take a closer look, we proceed with an $L$-layer MLP for its ease of presentation, despite the fact that our definition of feature complexity is architecture agnostic and our proposed method could be applied to arbitrary architectures. For a MLP, the feature map $Z^l(\theta) \in \mathbb{R}^{d_l}$ contains $d_l$ features. We use $Z_i^l(\theta), i \in [1, d_l]$ to denote the $i$-feature. The $\theta^l$ corresponds to a weight matrix $W^l \in \mathbb{R}^{d_l \times d_{l-1}}$ and a bias vector $\mathbf{b}^l \in \mathbb{R}^{d_l}$ and we have $Z^l(\theta) = \sigma(W^l Z^{l-1}(\theta) + \mathbf{b})$, where $\sigma$ corresponds to the activation function. For a permutation $\pi$ on $\theta$, we have $Z_\pi^l(\theta) = P_l Z^l(\theta)$ for each $l \in [1, L]$, where $P_l \in \mathbb{R}^{d_l \times d_l}$ is a permutation matrix. The weight is also permuted as $W_\pi^l = P_l W^l P_{l-1}^\top$ and $\mathbf{b}_\pi^l = P_l \mathbf{b}^l$, which make sure that $\forall \mathbf{x}, f(\theta, \mathbf{x}) = f(\pi(\theta), \mathbf{x})$.

### 2.2 CATEGORY THEORY VIEW OF NEURAL NETWORKS

We introduce basic and necessary concepts in category theory which is a branch of mathematics. For more comprehensive introduction please refer to (Mac Lane, 2013; Adámek et al., 1990).

**Category**: A category $C$ is a graph with a set of objects and arrows (or "morphisms"). For an arrow $f \in \mathbf{A}$ from object $a$ to $b$, it is written as $f : a \to b$ where $a$ is the domain of $f$ written as $\mathrm{dom} f = a$ and $b$ is the codomain of $f$ written as $\mathrm{cod} f = b$. A category also has two additional operations: *Identity*: For each object $a$ there exists an arrow $id_a : a \to a$. *Composition*: For each pair of arrows $< f, g >$ with $\mathrm{dom} g = \mathrm{cod} f$, there is an arrow $g \circ f : \mathrm{dom} f \to \mathrm{cod} g$ called their composite.

For the set of arrows from $b$ to $c$ in category $C$, it is written as $Hom_C(a, b)$ called "hom-set".

**Functor**: A functor is the morphism between categories. For two category $C$ and $B$, a functor $T : C \to B$ consists two suitably related functions: the object function, which assigns each object $c$ of $C$ an object $Tc$ of $B$ and the arrow function which assigns each arrow $f : c \to c'$ of $C$ an arrow $Tf : Tc \to Tc'$ of $B$, in such way that:

$$T(id_c) = id_{Tc}, \quad T(g \circ f) = Tg \circ Tf \tag{1}$$

It means the mappings defined by the functor $T$ preserves the structure of the category $C$.

**Natural transformation**: Given two functor $S, T : C \to B$, a natrual transformation $\tau : S \to T$ is a function assigns each object $c$ of $C$ an arrow $\tau_c : Sc \to Tc$ of $B$, in such way that every arrow $f : c \to c'$ of $C$ yields a commutative graph:

$$
\begin{array}{ccc}
Sc & \xrightarrow{\tau_c} & Tc \\
Sf \downarrow & & \downarrow Tf \\
Sc' & \xrightarrow{\tau_{c'}} & Tc'
\end{array}
\tag{2}
$$

Here commutative graph means that different paths between two objects yield, by composition, an equal arrow between the two objects, such that $Tf(\tau_c(Sc)) = \tau_{c'}(Sf(Sc))$.

For a neural network structure $f(\cdot, \cdot)$, it could be abstracted into a category $\mathcal{F}$ where objects are the shape of feature (feature map) and arrows are the type of transformation applied to the features, *e.g.* linear transformation, convolution, attention *etc.*. For an identity arrow, it simply applies no transformation on the feature while the composite of arrows is the composite of the corresponding transformations. Take a simple $L$-layer neural network structure for instance, it could be abstracted as the category depicted in Eq. 3, where each object corresponds to the shape of the input, feature maps and output while each arrow corresponds to the type of transformation applied to the feature (we omit identity circle and composites of arrows for simplicity).

$$\mathbf{x} \xrightarrow{f^1} Z^1 \xrightarrow{f^2} Z^2 \xrightarrow{f^3} \dots \xrightarrow{f^L} f(\cdot, x). \tag{3}$$

| Terms in category theory | Corresponding specific concept in our definition |
|---|---|
| Category | Neural network structure. |
| Object | Shape of feature maps including input and output (*e.g.* a tensor of shape $3 \times 224 \times 224$). |
| Arrow (Morphism) | Type of layers (transformations applied on the feature maps) (*e.g.* linear layer, convolution layer, *etc.*) |
| Functor | Parameterization of neural network Mapping each arrow (type of layer) to parameterized layer. |
| Natural transformation | There exist a natural transformation iff two neural network learn Functionally Equivalent Features. |

Table 1: Terms in category theory and their embodiment in our definitions about neural networks.

In the context of neural network structure, the training process aims to discover suitable parameters. In other words, it seeks to find a functor $T : \mathcal{F} \to \mathcal{P}$, where $\mathcal{P}$ represents a category describing the shapes of features, and its arrows encompass all possible transformations with specific parameters. For example, consider a network $f(\theta, \cdot)$ with parameter $\theta$, the corresponding functor $T_\theta$ maps each arrow $f^l(\cdot, \cdot)$ to $f^l(\theta^l, \cdot)$ with specific parameters $\theta^l$. To paint a clearer picture, as an analogy, we could think of the category $\mathcal{F}$ as a class in object-oriented programming language and think of the functor $T$ as creating an object instantiating the corresponding class. In Table 1, we list the terms in category theory that used in our definition and the corresponding concepts about neural networks.

## 2.3 DEFINITION OF FUNCTIONALLY EQUIVALENT FEATURE FROM THE CATEGORY VIEW

Through the lens of category theory, as introduced in Sec. 2.2, we further define functionally equivalent feature using natural transformation. Consider a natural isomorphism $\tau$ between two functors $T_{\theta_a}, T_{\theta_b} : \mathcal{F} \to \mathcal{P}$ satisfying that for each object $z \in \mathcal{F}$, the transform $\tau_z : T_{\theta_a} z \to T_{\theta_b} z$ is an invertible linear transformation. Natural transformations require naturality, ensuring that each arrow in category $\mathcal{F}$ results in a commutative graph in category $\mathcal{P}$, as described in Eq. 2. Therefore, natural isomorphisms between parameters are non-trivial.

**Definition 2.1.** **[Functionally Equivalent Features]** For a $L$-layer neural network $f(\cdot, \cdot)$ and two different parameter $\theta_a$ and $\theta_b$, if there is a natural isomorphism between functor $T_{\theta_a}$ and $T_{\theta_b}$ then model $f(\theta_a, \cdot)$ and model $f(\theta_b, \cdot)$ have functionally equivalent features such that

$$\forall \mathbf{x} \in \mathcal{D}, \forall l \in [1, L-1], \ \tau_{Z^{l+1}}(Z^{l+1}(\theta_b, \mathbf{x})) = f^{l+1}\left(\theta_a^{l+1}, \tau_{Z^l}(Z^l(\theta_b, \mathbf{x}))\right). \quad (4)$$

where $\tau_{Z^{l+1}}$ and $\tau_{Z^l}$ represents the invertible linear transformation defined by the natural isomorphism and $Z^l(\theta_b, \mathbf{x}) \in \mathbb{R}^{d_l}$ is the feature at the $l$-th layer.

According to Definition 2.1, natural isomorphisms between networks indicate the features modeled by two networks are functionally equivalent *i.e.* we could replace any feature maps of one model with the feature map from the other model through invertible linear transformation. Since it is a strong condition, simple transformations between different networks are unlikely to exist.

Specifically, we notice linear mode connectivity (LMC) (Frankle et al., 2020), an empirical phenomenon that have drawn extensive attention. It says that there may exist a linear path between two different neural networks such that along the path, the loss is nearly constant. Various methods (Entezari et al., 2022; Ainsworth et al., 2023; Liu et al., 2022) have been proposed to find networks satisfying LMC while recently a stronger notion of linear connectivity called Layerwise Linear Feature Connectivity (LLFC) was observed coexist with LMC (Zhou et al., 2023). We show that LMC actually indicates the functional equivalence between features.

**Proposition 2.2.** **[LMC indicates functionally equivalent features]** *(Proof in Appendix C). For two different parameter $\theta_a$ and $\theta_b$, if there is a permutation $\pi$ such that $\theta_a$ and $\pi(\theta_b)$ satisfy LMC, then there exists a natural isomorphism $\tau$ between functor $T_{\theta_a}$ and $T_{\theta_b}$.*

Proposition 2.2 provides an interpretation of linear mode connectivity from category theory perspective such that there always exists a natural isomorphism between two networks satisfying LMC, which indicates the two networks have functionally equivalent features. Note that Entezari et al. (2022); Ainsworth et al. (2023); Liu et al. (2022) showed that different networks can be linearly connected after permutation, therefore we assume each $\tau_z$ to be permutation in the following, which means establishing a one to one correspondence between the features of the two networks.

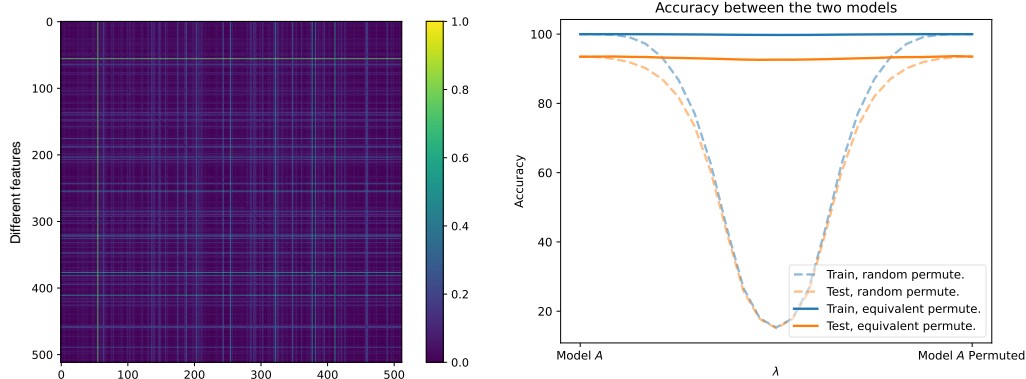

(a) Weight matching distance between different features (normalized).

(b) Testing accuracy on CIFAR10 of a model interpolating between a VGG16 and the same model after permutation.

Figure 2: The empirical evidences that functionally equivalent features exist. (a): The heatmap of the distances between different features from the last convolution layer of VGG16 on CIFAR10. (b):The test accuracy on CIFAR10 of a model interpolating between a VGG16 and the same model after permutation on each layer. The solid lines are the result of permutation on functionally equivalent features while the dashed lines are the result of random permutation on the same number of features.

## 2.4 DEFINITION OF FEATURE COMPLEXITY FROM THE EQUIVALENT FEATURE

Based on the definition of functionally equivalent feature, we could further define the feature complexity. Consider natural transformations mapping the functor $T_\theta$ to itself. The natural transformation that maps each arrow to itself must exist. In the case of other natural transformations, they map $Z^l(\theta)$ to $Z^l(\theta)P$ where $P \in \mathbb{R}^{d_l \times d_l}$ is a non-identity permutation matrix, which means mapping feature $Z_i^l(\theta)$ to $Z_j^l(\theta)$ with $i \neq j, i, j \in [1, d_l]$. This approach allows us to establish a partial order among features. For two features $Z_i^l(\theta), Z_j^l(\theta), i, j \in [1, d_l]$ at $l$-th layer of network $f(\theta, \cdot)$, if there exists a simple transformation between $T_\theta$ to itself that maps $Z_i^l(\theta)$ to $Z_j^l(\theta)$, we say $Z_i^l(\theta) \leq Z_j^l(\theta)$. It indicates that feature $Z_j^l(\theta)$ covers the feature represented in $Z_i^l(\theta)$ and could be used to replace $Z_i^l(\theta)$. It is obvious that if $Z_i^l(\theta)$ and $Z_j^l(\theta)$ is comparable then $Z_i^l(\theta) = Z_j^l(\theta)$.

**Theorem 2.3.** *[Feature duality] (Proof in Appendix C). For a L-layer neural network $f(\theta, \cdot)$, there are multiple natural isomorphisms between $T_\theta$ to itself if and only if*

$$\exists l \in [1, L], \exists i, j \in [1, d_l], s.t. \quad Z_i^l(\theta) = Z_j^l(\theta), i \neq j. \tag{5}$$

Finally, we formally define feature complexity based on the partial ordering between features.

**Definition 2.4. [Feature complexity]** Given a $L$-layer neural network $f(\theta, \cdot)$, the feature at $l$-th layer compose a poset $\{Z_n^l(\theta)|n \in [1, d_l]\}$. The maximum number of features that are not equivalent *i.e.* the width of the corresponding poset is defined as the feature complexity at $l$-th layer.

## 3 MEASURING NEURAL NETWORK FEATURE COMPLEXITY

In this section, we propose an algorithm to empirically measure the complexity of the features by finding equivalent features with weight matching and merging the equivalent features. Firstly, we devise and introduce two components to find and merge equivalent features.

**Feature weight matching:** Drawing inspiration from the weight matching method used to identify permutations in linear mode connectivity literature, we propose matching the weights corresponding to each feature. Denote $W_{[i,:]}^l$ as the $i$-th row of $W^l$ and $W_{[:,j]}^l$ as the $j$-th column of $W^l$. For two features $Z_m^l, Z_n^l, m, n \in [1, d_l]$, the weight distance between the two features is defined as:

$$D_{mn}^l = \|W_{[m,:]}^l - W_{[n,:]}^l\|^2 + \|W_{[:,m]}^{l+1} - W_{[:,n]}^{l+1}\|^2 \tag{6}$$

When matching the weights, we consider the layers such as linear fully connected layers or convolution layers at the $l$-th layer, while ignoring the activation and normalization layers in the network.

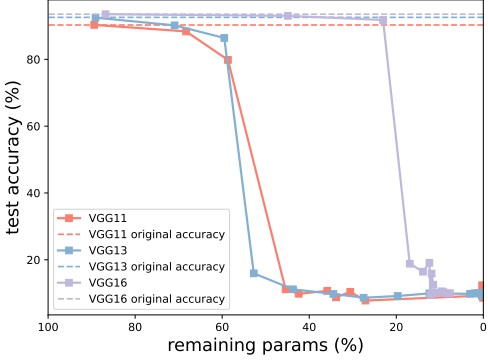 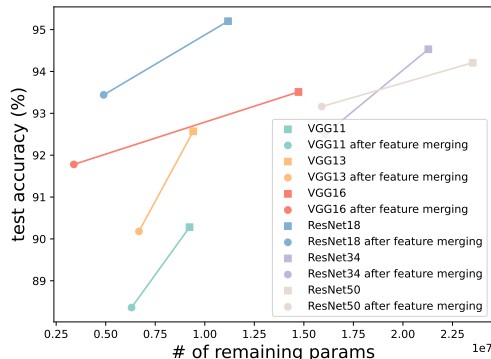

(a) Testing accuracy to the percentage of remaining parameters on VGG networks

(b) Testing accuracy to the number of remaining parameters on VGG and ResNet

Figure 3: Results for iterative feature merging on CIFAR10. (a): Testing accuracy to the percentage of remaining parameters. Each dot corresponds to a different hyper-parameter $\beta$ in iterative feature merging. (b): Testing accuracy to the number of parameters. We conduct a grid search on $\beta$ and choose the largest $\beta$ with testing accuracy larger than $95\%$ of the testing accuracy before merging.

**Feature merging:** Given two functionally equivalent features $Z_m^l, Z_n^l$ at the $l$-th layer, where $m, n \in [1, d_l]$, To obtain the merged weight at the $l$-th layer $W'^l \in \mathbb{R}^{(d_l-1) \times d_{l-1}}$, the row of weight matrix $W'^l$ corresponding to the merged feature is calculated by:

$$W'^l_{merged} = W^l_{[m,:]} + W^l_{[n,:]} \tag{7}$$

The other $d_l - 2$ rows of $W'^l$ correspond to the $d_l - 2$ rows in $W'^l$ such as $W^l_{[i,:]}, i \neq m, n$.

Therefore the merged feature at $l$-th layer becomes $Z'^l \in \mathbb{R}^{d_l - 1}$ where the merged feature $Z'^l_{d_l-1} = Z_m^l + Z_n^l$ and the other $d_l - 2$ features correspond to $Z_i^l, i \neq m, n$.

To process the $Z'^l$, we adjust the weight at $l + 1$-th layer. For $W'^{(l+1)} \in \mathbb{R}^{d_{l+1} \times d_l - 1}$, the column of $W'^{(l+1)}$ corresponding to the merged feature is:

$$W'^{(l+1)}_{merged} = mean(W^{l+1}_{[:,m]}, W^{l+1}_{[:,n]}) \tag{8}$$

The other $d_l - 2$ columns of the weight $W'^{(l+1)}$ corresponds to the $d_l - 2$ columns $W^l_{[:,j]}, j \neq m, n$.

Using this method, we could merge features to obtain an approximately functionally equivalent neural network. Note that when merging several features the process is similar to Eq. 7 and Eq. 8.

**Iterative Feature Merging with Weight Matching:** Composing the two components proposed above, we get the algorithm to measure the feature complexity. We iteratively merge the features in each layer $l$ with the smallest weight distance defined in Eq. 6 until

$$\min_{m,n \in [1,d_l], m \neq n} D_{mn}^l > \beta \max_{m,n \in [1,d_l], m \neq n} D_{mn}^l \tag{9}$$

With hyper-parameter $\beta$, $D_{mn}^l$ corresponds to the weight distance as defined in Eq. 6.

To get the mean value in Eq. 8, we also keep track of the number of features merged into one feature. Consider merging two feature $Z_m^l$ and $Z_n^l$ where $Z_m^l$ is the merged feature of $N_m$ features and $Z_n^l$ is the merged feature of $N_n$ features, then we have:

$$W'^{(l+1)}_{merged} = \left( N_{m_{min}} W^{l+1}_{[:,m_{min}]} + N_{n_{min}} W^{l+1}_{[:,n_{min}]} \right) / \left( N_{m_{min}} + N_{n_{min}} \right) \tag{10}$$

The detailed algorithm for Iterative Feature Merging (IFM) is represented in Algorithm 1.

## 4 EXPERIMENTS

In Sec. 4.1, we empirically verify the existence of functionally equivalent features. In Sec. 4.2, we show that the iterative feature merging could greatly reduce the number of parameters while

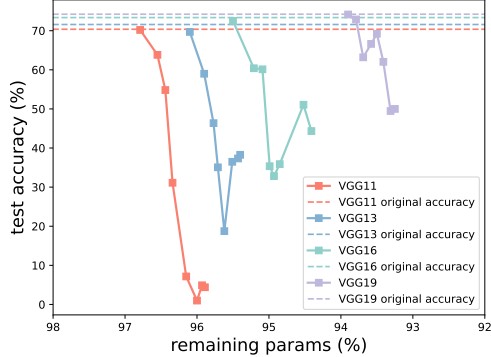 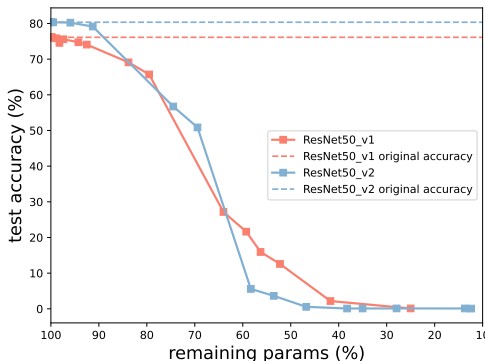

(a) Testing accuracy to the percentage of remaining parameters on VGG networks

(b) Testing accuracy to the percentage of remaining parameters on two different ResNet50

Figure 4: Results of iterative feature merging on ImageNet. (a): Testing accuracy (top-1) in relation to the percentage of remaining parameters of VGGs after feature merging with different $\beta$. (b): Testing accuracy (top-1) to the percentage of remaining parameters of ResNet50 after feature merging. "ResNet50_v1" and "ResNet50_v2" corresponds to the two different checkpoints in torchvision.

maintaining the performance without fine-tuning. In Sec. 4.3, we provide more empirical results regarding the iterative feature merging. We conducted experiments using VGG and ResNet models on CIFAR10 and ImageNet, respectively. For more details, refer to Appendix D.

## 4.1 EXISTENCE OF EQUIVALENT FEATURES

In this section, we empirically show that the functionally equivalent feature as defined in Sec. 2.3 actually exist in vanilla networks. We conduct experiments on VGG16 trained on CIFAR10 with SGD optimizer for 150 epochs. For details, refer to Appendix D.

As shown in Fig. 2(a), we visualize the normalized distances between 512 different features from the last convolution layer of the VGG16. The minimum distance between two features is $2e - 15$. For most features, the distance between them is relatively low. Similar results were observed for other layers. For more results please refer to Appendix E.

For a model $f(\theta, \cdot)$, we further consider the permutation $\pi$ that swaps the functionally equivalent features found by the feature weight matching introduced in Sec. 3. Similar to the metric used to identify LMC, we further interpolate between the parameter $\theta$ and the parameter after permutation $\pi(\theta)$ and test the accuracy of model $f((1 - \lambda)\theta + \lambda\pi(\theta), \cdot)$. The results are shown in Fig. 2(b) where solid lines correspond to the permutation swapping equivalent features and dashed lines correspond to random permutation swapping the same number of features for comparison. The results shows that interpolating between functionally equivalent features does not affect the performance.

## 4.2 ITERATIVE FEATURE MERGING

In this section, we present the results of iterative feature merging with VGG (Simonyan & Zisserman, 2015) and ResNet (He et al., 2016) on CIFAR10 (Krizhevsky et al., 2009) and ImageNet (Deng et al., 2009). We trained models on CIFAR10 and use pretrained checkpoints on ImageNet provided by torchvision (maintainers & contributors, 2016). For details, refer to Appendix D.

**Iterative Feature Merging on CIFAR10**

By varying $\beta$ in Algorithm 1, we could observe the change of testing accuracy as different number of features are merged. The results for VGG networks on CIFAR10 are shown in Fig. 3(a), with the x-axis representing the percentage of remaining parameters and the y-axis representing testing accuracy. **It is clear that the larger the network is, the more equivalent features could be merged before significantly affecting the performance.** Specifically, for VGG16, we could reduce the number of parameters to 23.03% while keeping the testing accuracy at 91.78% which is relatively close to the original testing accuracy at 93.51%.

We further demonstrate the relationship between testing accuracy and the number of network parameters in Fig. 3(b). Here we apply grid search on $\beta$ and choose the largest $\beta$ with the testing

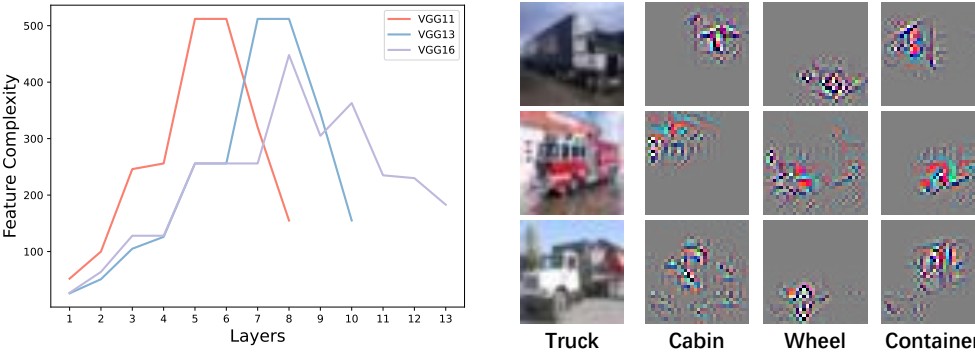

(a) Feature complexity at each layer of VGGs     (b) Guided backpropagation results

Figure 5: Empirical results regarding feature complexity. (a): The plot of feature complexity at each layer of VGG networks that is trained on CIFAR10. (b): The results of guided backpropagation of three different sets of equivalent features in the VGG16 on three different images from CIFAR10.

accuracy larger than $95\%$ of the testing accuracy before feature merging. **With iterative feature merging, larger models (*e.g.* VGG16 and ResNet50) could reduce more parameter with a smaller decrease on testing accuracy.** Note that we also find ResNet18 outperforming ResNet34 and ResNet50 on CIFAR10. For the reason of this phenomenon, we believe that the capacity of ResNet18 is already enough to deal with the simple classification task on CIFAR10, *e.g.* we could reduce up to $60\%$ of the parameters of ResNet-18 without significantly affecting the testing accuracy.

**Iterative Feature Merging on ImageNet**

We find that neural networks learn more complex features to solve the more difficult task, *i.e.* image classification on ImageNet. As shown in Fig. 4, iterative feature merging only reduce approximately $5\%$ parameters for VGG and approximately $10\%$ parameters for ResNet50.

We observe the same trend as on CIFAR10 that the larger the network is, the more equivalent features could be merged, as shown in Fig. 4(a). For different parameters with the same network structure, we conduct experiments on two different parameters of ResNet50 in torchvision, where "ResNet50_v2" achieves higher top-1 accuracy than "ResNet50_v1" ($80.86\%$ to $76.13\%$) due to different training hyper-parameters. As shown in Fig. 4(b), the testing accuracy of ResNet50_v2 decreases faster than the testing accuracy of ResNet50_v1 as more features are merged. It indicates that ResNet50_v2 learns more diverse features, which may explain the superior performance of ResNet50_v2.

**The potential of IFM for pruning**

IFM merges functionally equivalent features, which is similar to the channel pruning (Wen et al., 2016; Ye et al., 2018) that remove channels to reduce computational cost. Note that conventional pruning methods require fine-tuning the pruned model while our proposed method does not. In Fig. 6, we compare our IFM with a recently SOTA pruning method "INN" (Solodskikh et al., 2023) which also does not require the fine-tuning. Specifically, INN requires changing the training procedure while our IFM can be directly applied to vanilla pre-trained models with high efficiency. See Appendix E for the time complexity analysis. However, it is worth noting that the pruning effect of IFM may be limited on complex tasks such as image classification on ImageNet.

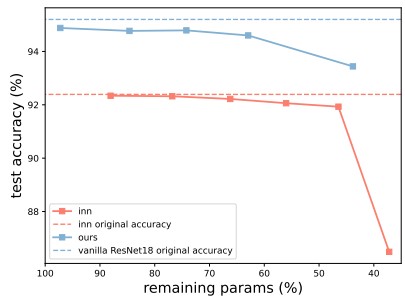

Figure 6: Pruning results of ResNet18 on CIFAR10. The INN experiments are conducted with the released official code.

### 4.3 EMPIRICAL RESULTS ON FEATURE COMPLEXITY

Fig. 5(a) shows the feature complexity at each layer of the VGG networks trained on CIFAR10. **We find that the feature complexity is increasing at the first several layers and decreasing at the last several layers while the maximum feature complexity is reached in the middle.** Several possible reasons could be used to explain this phenomenon, *e.g.* the neural network starts to forget

features or many simple features compose complex features at the last several layers. We leave the explanation of this phenomenon for future work.

In Fig. 5(b), we visualize the guided backpropagation of various features in VGG16. Guided backpropagation (Springenberg et al., 2015) is a method to visualize the part of the image that activate neurons in the neural network by applying ReLU on the gradient through backpropagation. Here we visualize the guided backpropagation of a group of functionally equivalent features by setting the gradient of other features to be zero. As shown in Fig. 5(b), a group of functionally equivalent features capture the similar features on different images. The three different groups of functionally equivalent features we demonstrate capture the cabin, wheel and container of trucks respectively. **It shows the possibility to align semantics to a group of equivalent features learned by the neural network for classification.** Please refer to Appendix E for more results.

## 5 RELATED WORKS

**Model Similarity.** To harness the behaviour of neural networks in train/test, efforts have been made especially for features (representations) learned by networks: the transferability of features (Yosinski et al., 2014), the training dynamics (Morcos et al., 2018), the effect of network width and depth on the learned feature (Nguyen et al., 2021). Among them, measuring the feature similarity (representational similarity) has become an important issue, with various metrics to measure the feature similarity including those based on canonical correlation analysis (Raghu et al., 2017; Morcos et al., 2018); the measures based on alignment (Li et al., 2016; Williams et al., 2021); the measures based on representational similarity matrix (Shahbazi et al., 2021; Tang et al., 2020) and the measures based on topology (Barannikov et al., 2022). Based on various feature similarity measures, previous works have shown that different networks with similar performance learn similar features (Li et al., 2016; Wang et al., 2018) and there are similar features within a network with increased width/depth (Nguyen et al., 2021). For measuring similarity between models, another focus is functional similarity (Madani et al., 2004; Bansal et al., 2021; Bhojanapalli et al., 2021; Klabunde et al., 2023) *i.e.* measuring the similarity of outputs. However, Ding et al. (2021); Hayne et al. (2022) show that functional similarity and feature similarity are not necessarily correlated such that similar features can yield different outputs while similar outputs can be obtained from different features. In this paper, we propose functionally equivalent feature, extending the concept of equivalent feature in feature similarity literature and functional measures in functional similarity literature.

**Linear Mode Connectivity.** Freeman & Bruna (2017); Draxler et al. (2018); Garipov et al. (2018) observed Mode Connectivity, i.e., different well-trained models can be connected through a nonlinear path of nearly constant loss. Frankle et al. (2020) first proposed the notion of Linear Mode Connectivity (LMC), where models are connected through linear path of constant loss. Frankle et al. (2020) observed LMC for networks that are jointly trained for a short amount of time before going through independent training (referred as the spawning method). Later, Entezari et al. (2022); Ainsworth et al. (2023); Liu et al. (2022) showed that even independently trained networks can be linearly connected when permutation invariance is taken into account. In particular, Ainsworth et al. (2023) utilized the permutation invariance to align the neurons of two neural networks and formulate the neuron alignment problem as a bipartite graph matching problem. More recently, Zhou et al. (2023) discovered a stronger notion of linear connectivity, called Layerwise Linear Feature Connectivity (LLFC), which indicates that the feature maps of every layer in different trained networks are also linearly connected, and demonstrated the co-occurrence of LLFC and LMC.

**Category Theory in Machine Learning.** Category theory have been used in machine learning (Shiebler et al., 2021) for various topics including backpropagation (Fong et al., 2019), categorical probabilities (Fritz, 2020), conditional independence (Mahadevan, 2022a), supervised learning (Harris, 2019), reinforcement learning (Mahadevan, 2022b) and so on. More recently, Yuan (2023) provides an analysis upon the power of perfect foundation models using category theory.

## 6 CONCLUSION

We have defined functionally equivalent features and introduce the concept of feature complexity, as learned by the network. To measure the feature complexity, we propose an efficient algorithm called Iterative Feature Merging (IFM). Experiments on CIFAR10 and ImageNet show its efficiency and inspire insights regarding the feature complexity. Our IFM also shows potential in pruning pre-trained models without fine-tuning. See more discussion in Appendix A.

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

## A  DISCUSSION ON LIMITATION AND FUTURE WORK

Since we do not proactively modify the structure and training procedure of the neural network, when facing difficult tasks (ImageNet classification), the number of parameters that could be reduced by IFM is limited. For better pruning performance, designing network structure and training algorithms to learn a more compact representation could be a promising direction, which we leave for future researches. Though our proposed definition and method are structure and data agnostic, another major limitation is that we only conduct experiments for image classification task. More experiments could be done regarding different network structures and different tasks. We hope that our proposed feature complexity will inspire future research for a better understanding and enhancement of neural network behavior.

## B  ALGORITHM FOR ITERATIVE FEATURE MERGING

In this section, we present the detailed algorithm for iterative feature learning (IFM) in Algorithm 1.

---

**Algorithm 1** Iterative Feature Merging with Weight Matching

---

**Input:** Parameter $\theta$ for a $L$-layer neural network $f(\cdot, \cdot)$, where the weight at the $l$-th layer is $W^l \in \mathbb{R}^{d_l \times d_{l-1}}$; hyperparameter $\beta$
**for** $l \in [1, L]$ **do**
  **while** True **do**
    **for** $m \in [1, d_l]$ **do**
      $N_m \leftarrow 1$
      **for** $n \in [1, d_l]$ **do**
        $D^l_{mn} \leftarrow \|W^l_{[m,:]} - W^l_{[n,:]}\|^2 + \|W^{l+1}_{[:,m]} - W^{l+1}_{[:,n]}\|^2$
      **end for**
    **end for**
    **if** $\min_{m \neq n} D^l_{mn} > \beta \max_{m \neq n} D^l_{mn}$ **then**
      break
    **else**
      $m_{min}, n_{min} \leftarrow \arg\min_{m,n,m \neq n} D^l_{mn}$
      $W'^l_{merged} \leftarrow W^l_{[m_{min},:]} + W^l_{[n_{min},:]}$
      $W'^{(l+1)}_{merged} \leftarrow (N_{m_{min}} W^{l+1}_{[:,m_{min}]} + N_{n_{min}} W^{l+1}_{[:,n_{min}]})/(N_{m_{min}} + N_{n_{min}})$
      $N_{merged} \leftarrow N_{m_{min}} + N_{n_{min}}$
    **end if**
  **end while**
**end for**

---

## C  PROOFS

### C.1  PROOF FOR PROPOSITION 2.2

Let us first formally define linear mode connectivity.

**Definition C.1. [Linear Mode Connectivity]** Given a dataset $\mathcal{D}$ and two neural networks of the same structure and different parameters $\theta_a$ and $\theta_b$ with similar loss $\mathcal{L}_\mathcal{D}(\theta_a) \approx \mathcal{L}_\mathcal{D}(\theta_b)$, the two networks are linearly connected if

$$\mathcal{L}_\mathcal{D}(\theta_a) \approx \mathcal{L}_\mathcal{D}(\theta_b) \approx \mathcal{L}_\mathcal{D}(\alpha\theta_a + (1-\alpha)\theta_b); \quad \forall\alpha \in [0,1]. \tag{11}$$

Beyond LMC, the linear layer-wise feature connectivity (LLFC) was observed coexist with LMC.

**Definition C.2. [Layerwise Linear Feature Connectivity]** Given a dataset $\mathcal{D}$ and two $L$-layer neural networks of the same structure and different parameters $\theta_a$ and $\theta_b$. They satisfy LLFC if

$$\forall\mathbf{x} \in \mathcal{D}, \forall l \in [1, L], \forall\alpha \in [0, 1], Z^l(\alpha\theta_a + (1-\alpha)\theta_b, \mathbf{x}) = \alpha Z^l(\theta_a, \mathbf{x}) + (1-\alpha)Z^l(\theta_b, \mathbf{x}). \tag{12}$$

*Proof.* [**Proof for Proposition** 2.2] Consider two parameter $\theta_a$ and $\theta_b$ satisfying LMC, according to Eq. 12 we have

$$f^l\left(\alpha\theta_a + (1-\alpha)\theta_b, \alpha Z^{l-1}(\theta_a, \mathbf{x}) + (1-\alpha)Z^{l-1}(\theta_b), \mathbf{x}\right) = \alpha f^l\left(\theta_a, Z^{l-1}(\theta_a, \mathbf{x})\right) + (1-\alpha)f^l\left(\theta_b, Z^{l-1}(\theta_b, \mathbf{x})\right)$$
$$(13)$$

When $f^l$ is activation layer such as ReLu, we have

$$\sigma\left(\alpha Z^{l-1}(\theta_a, \mathbf{x}) + (1-\alpha)Z^{l-1}(\theta_b), \mathbf{x}\right) = \alpha\sigma\left(\theta_a, Z^{l-1}(\theta_a, \mathbf{x})\right) + (1-\alpha)\sigma\left(\theta_b, Z^{l-1}(\theta_b, \mathbf{x})\right)$$

Here $\sigma(\cdot)$ represent the non-linear function, therefore we have

$$Z^l(\theta_b, \mathbf{x}) = f^l(\theta_a, Z^{l-1}(\theta_b, \mathbf{x}))$$

For layers that perform linear transformation such as linear layer, we can write Eq. 13 in matrix form:

$$\left(\alpha W^l_{\theta_a} + (1-\alpha)W^l_{\theta_b}\right)\left(\alpha Z^{l-1}(\theta_a, \mathbf{x}) + (1-\alpha)Z^{l-1}(\theta_b, \mathbf{x})\right) = \alpha W^l_{\theta_a} Z^{l-1}(\theta_a, \mathbf{x}) + (1-\alpha)W^l_{\theta_b} Z^{l-1}(\theta_b, \mathbf{x})$$

Then we have

$$W^l_{\theta_a} Z^{l-1}(\theta_a, \mathbf{x}) + W^l_{\theta_b} Z^{l-1}(\theta_b, \mathbf{x}) = W^l_{\theta_a} Z^{l-1}(\theta_b, \mathbf{x}) + W^l_{\theta_b} Z^{l-1}(\theta_a, \mathbf{x})$$

Derive it, we get

$$\left(W^l_{\theta_a} - W^l_{\theta_b}\right)\left(Z^{l-1}(\theta_a, \mathbf{x}) - Z^{l-1}(\theta_b, \mathbf{x})\right) = 0$$

Therefore, we have

$$Z^l(\theta_b, \mathbf{x}) = f^l(\theta_a, Z^{l-1}(\theta_b, \mathbf{x}))$$

Similarly, when $\theta_a$ and $\pi(\theta_b)$ satisfy LMC, we have

$$P^l Z^l(\theta_b, \mathbf{x}) = Z^l(\pi(\theta_b), \mathbf{x}) = f^l(\theta_a, Z^{l-1}(\pi(\theta_b), \mathbf{x})) = f^l(\theta_a, P^{l-1}Z^{l-1}(\theta_b, \mathbf{x})) \qquad (14)$$

Here $P^l$ and $P^{l-1}$ is the permutation matrix defined by $\pi$. Therefore, we could simply define a corresponding natural isomorphism such that each $\tau_{z^l}$ is $P^l$ ☐

## C.2 PROOF FOR THEOREM 2.3

*Proof.* According to the partial order between features defined in Sec. 2.4, when there is more than one natural isomorphisms between $T_\theta$ to itself then we have

$$\exists l \in [1, L], \exists i, j \in [1, d_l], s.t. \quad Z^l_i(\theta) \leq Z^l_j(\theta), i \neq j. \qquad (15)$$

Next, we need to prove that

$$Z^l_i(\theta) = Z^l_j(\theta) \qquad (16)$$

Consider the poset $\{Z^l_n(\theta) \mid n \in [1, d_l]\}$, since the natural isomorphism defines a one-to-one correspondence between the features, the objects in the same chain are equal. Otherwise, the maximal element $Z^l_{max\_S}$ in a chain $S$ must have

$$Z^l_{max\_S} \leq Z^l_k, \quad Z^l_k \notin S \qquad (17)$$

which contradict to the statement that $Z^l_{max\_S}$ is in chain $S$. Therefore we have

$$\exists l \in [1, L], \exists i, j \in [1, d_l], s.t. \quad Z^l_i(\theta) = Z^l_j(\theta), i \neq j. \qquad (18)$$

☐

## C.3 DETAILED INTRODUCTION OF THE FORMULATION WITH CATEGORY THEORY

**The Category Corresponding to Model Structure.** As we abstractly present the model structure with a category, it could be depicted as follows (we omit identity circle and composites of arrows for simplicity).

$$\mathbf{x} \xrightarrow{f^1} Z^1 \xrightarrow{f^2} Z^2 \xrightarrow{f^3} \dots \xrightarrow{f^L} f(\cdot, x).$$

The objects ($\mathbf{x}$, $Z^1$, *etc.*) in the category are the shape of the feature maps (including inputs and output). For example, for neural network structures designed for ImageNet, the $\mathbf{x}$ here is a tensor of shape $3 \times 224 \times 224$. Arrows between the objects indicate the type of transformations applied on the feature maps. For example, the first layer $f^1$ for VGG is a convolution with a $3 \times 3$ convolution kernel. For it to be a category, according to the definition of category, two additional conditions should be satisfied:

- *Identity*: For each object $a$ there exists an arrow $id_a : a \rightarrow a$.

- *Composition*: For each pair of arrows $< f, g >$ with $\mathrm{dom}g = \mathrm{cod}f$, there is an arrow $g \circ f : \mathrm{dom}f \rightarrow \mathrm{cod}g$ called their composite.

Identity condition could be easily satisfied where we add identity arrow to each object which means we conduct no transformation to the feature maps. For composition, we add composition of arrows corresponds to composition of transformations (e.g. $f^1 \circ f^2$ means we firstly apply transformations in the first layer on $\mathbf{x}$ then apply transformations in the second layer and produce $Z^2$).

**The Functor Corresponding to Parameterization of Neural Network.** Functors, by definition, map each object and arrow from on category to another category, while preserving the structure. For functors in our definition, it map the category of network structure to the category of tensors where objects are tensors while arrows are transformations between them. For a functor T, it maps each arrow (the type of a transformation) to a specific transformation with specific parameters.

## D  EXPERIMENT DETAILS

**Training details for models on CIFAR10** We train models with the same hyper-parameters. For each model, we train it using SGD with momentum at $0.9$ and weight decay at $1e-4$ for 150 epochs. The initial learning rate is at $0.1$ and we reduce learning rate at $80$ and $120$ epoch by multiply it with $0.1$. For data augmentation, we only use random horizontal flip with probability set at $50\%$.

Note that the model structures are a little bit different to the structures on ImageNet. For VGGs, the number of layers for the classifier is reduced to $1$ instead of $3$. For ResNet, we apply the conventional change such that the convolution kernel size is set to be $3$ at the first layer.

**Checkpoint details for models on ImageNet** For each model, we use pretrained checkpoints on ImageNet provided by pytorch (maintainers & contributors, 2016). Note that there are two different checkpont for ResNet50 named "ResNet50_Weights.IMAGENET1K_V1" and "ResNet50_Weights.IMAGENET1K_V2"

**Details for Iterative Feature Merging** For IFM, the only hyper-parameter is the $\beta$ in Algorithm 1. We grid search it in the list [0.01, 0.03, 0.05, 0.07, 0.1, 0.12, 0.14, 0.15, 0.18, 0.2].

Note that for ResNet, we only merge the features of the last two block for better merging effect.

## E  ADDITIONAL RESULTS

### E.1  TIME COMPLEXITY OF IFM

The comutational cost of IFM is relative low since we only need to compute the weight matching distance between features, which is faster than conducting model inference. Table E.1 shows the time consumption of one iteration for different models on CIFAR10 and ImageNet where each result if average over 100 iteration. The result is conducted with one Geforce RTX 2080Ti.

Table 2: Time consumption for one iteration with different models on different dataset. The result is averaged over 100 iterations.

| dataset | model | time for one iteration (s) |
|---------|-------|----------------------------|
| CIFAR10 | VGG13 | $0.018 \pm 0.003$ |
| | VGG16 | $0.027 \pm 0.005$ |
| | ResNet18 | $0.015 \pm 0.003$ |
| | ResNet50 | $0.127 \pm 0.012$ |
| ImageNet | VGG13 | $0.802 \pm 0.053$ |
| | VGG16 | $0.814\pm : 0.054$ |
| | VGG19 | $0.822 \pm 0.045$ |
| | ResNet18 | $0.015 \pm 0.002$ |
| | ResNet50 | $0.127 \pm 0.010$ |

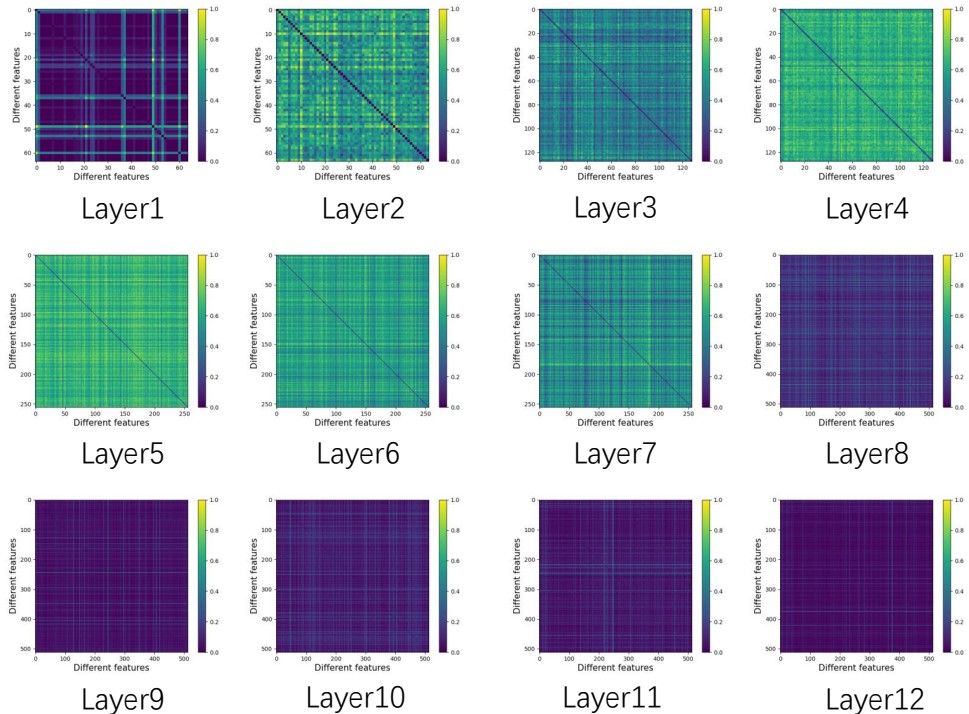

Figure 7: Distance weight between features at the first 12 layers of a VGG16 trained on CIFAR10.

### E.2 WEIGHT DISTANCE BETWEEN FEATURES

In Fig. 2.4, we present the weight distance between features at the last convolution layer of a VGG16 trained on CIFAR10. In this section, we present the result of the first 12 layers in Fig. E.2.

### E.3 GUIDED BACK-PROPAGATION RESULTS

In this section, we show more empirical results of the guided backpropagation of different groups of functionally equivalent features. In Fig. 5(b), we show that a group of features may corresponds to a semantic in a class. In Fig. 8, we further show that a group of functionally equivalent features may correspond to a semantic across different classes, where the specific group of features are activated by legs which is presented both on horses, cats and dogs. We also provide results in Fig. 9 that are similar to Fig. 5(b) where different semantics regarding boats are learned by groups of functionally equivalent features. Note that all these results are from features at the middle of VGG16, since the result of most features at the last layer cover the whole image. Future works may explore more on the different groups of functionally equivalent features.

## F FURTHER DISCUSSION: RELATIONSHIP BETWEEN IFM AND MODEL COMPRESSION METHODS

**Brief Introduction of Model Compression and Channel Pruning.** The increasing size of neural networks have motivated the standing research on network compression over the decade. To reduce the computation cost, various methods have been proposed: knowledge distillation (Hinton et al., 2015), quantization (Gong et al., 2014; Wu et al., 2016), low-rank factorization (Tai et al., 2015), *etc*. Specifically, network pruning (Hanson & Pratt, 1988) tend to reduce redundant parameters that are not sensitive to the performance. The pruning methods include unstructured pruning (LeCun et al., 1989; Hassibi et al., 1993; Srinivas & Babu, 2015) and structured pruning (Zhou et al., 2016; Wen et al., 2016; Ye et al., 2018). The unstructured pruning remove connections between neurons that

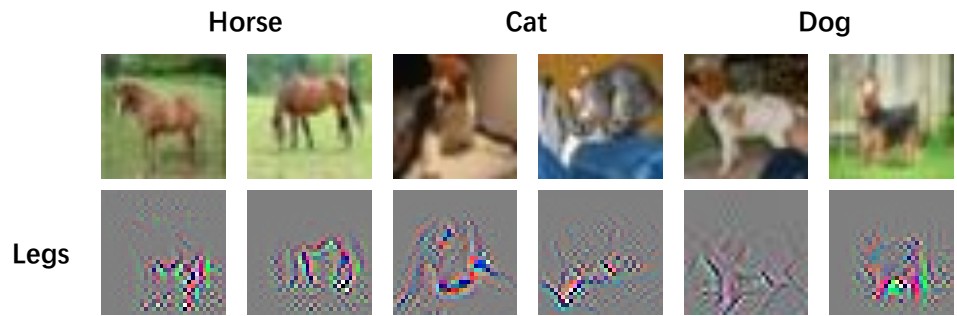

Figure 8: Guided backpropagation result of the VGG16 on CIFAR10. Here we show that a group of functionally equivalent features may corresponds to a semantic across several classes.

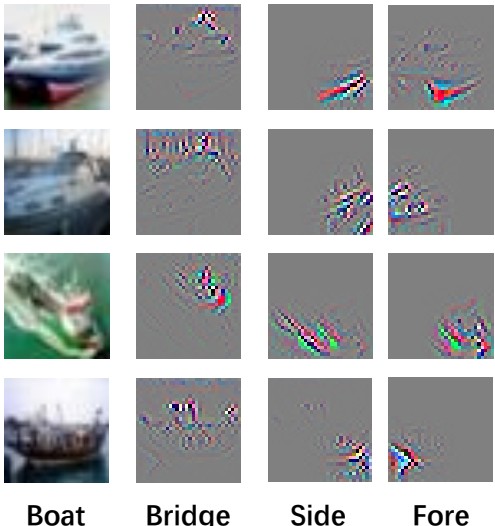

Figure 9: Guided backpropagation result of the VGG16 on CIFAR10. Here we show that a group of functionally equivalent features may corresponds to a semantic regarding boats.

Table 3: Comparison between popular pruning methods about whether they require access to data before and after pruning. Note that our method requires no data both before and after pruning.

| Method | access to data before pruning | training (fine-tuning) after pruning |
|---|---|---|
| Structured Sparsity Learning (SSL) (Wen et al., 2016) | ✓ | × |
| Iterative Magnitude Pruning(IMP) (Frankle & Carbin, 2019; Frankle et al., 2019) | ✓ | ✓ |
| SNIP (Lee et al., 2019) | ✓ | ✓ |
| Iterative Synaptic Flow Pruning (SynFlow) (Tanaka et al., 2020) | × | ✓ |
| Integral Neural Network (INN) (Solodskikh et al., 2023) | ✓ | × |
| Iterative Feature Merging (IFM) (Ours) | × | × |

the dimensionality of the feature map does not change. The structured pruning, on the other way, removes groups of weights (filters, channels, *etc.*) and reduces the dimension of the feature map.

**Difference between IFM and Channel Pruning Methods.** In general, the effect of IFM and channel pruning is similar: reducing the dimension of the feature map. The key difference is that IFM has different purpose compared with channel pruning methods. The proposed IFM aims at finding the most compact representation without significantly affect the performance while the purpose of channel pruning is better preserving the performance under certain computation budget. The empirical difference is that the proposed IFM does not require access to any data. Efforts have also been made to develop data agnostic prune methods. For pruning at initialization (PaI) (Wang et al., 2021), Tanaka et al. (2020) propose a data agnostic method to prune the network before training. For pruning after training (PaT), Solodskikh et al. (2023) propose a new training algorithm that does not require fine-tuning to prune the model after training. However, they either require the access to training (fine-tuning) procedure after pruning or before pruning. When considered as a prune method, to the best of our knowledge, the IFM is the first data-agnostic prune method not requiring any training or fine-tuning (as shown in Table 3). On the other hand, the compression ratio could not be precisely controlled in IFM. Therefore, fine-tuning might also be required after IFM for the proposed method to be an actual prune method.

