# OpenReview forum: "Going Beyond Neural Network Feature Similarity: The Network Feature Complexity and Its Interpretation Using Category Theory"
_ICLR.cc/2024/Conference — ICLR 2024 poster_

### Official Review · Reviewer_Po5f · 2023-10-27

**Soundness:** 3 good
**Presentation:** 3 good
**Contribution:** 3 good
**Rating:** 8
**Confidence:** 4

**Summary:**

The paper delves into the feature similarity and further proposes the feature equivalence, and finally feature complexity of a trained neural network. The authors also introduce the well-established math tool of category theory to elegantly estaiblish the theory of the introduced concepts as well as methods. Beyond new theoretical understanding, the authors further devise an iterative algorithm to achieve the computing of feature complexity based on which the functionally equivalent features can be found in a neural network such that pruning of neural networks can be fulfilled.

**Strengths:**

1) an important understanding to the trained neural networks by introducing the new concept of feature equivalence and feature complexity which go beyond the feature similarity.
2) an effective algorithm for computing the feature complexity and as a side-product can be used to prune the trained networks on the fly without access to the training dataset, neither with any iterations for finetuning which are needed in existing pruning methods.
3) the authors further draw a few interesting observations which are well interpretable that further consolidates the impact and soundness of this work.

**Weaknesses:**

The paper could be improved in some aspects: 1) due to the density of the new information in this paper, the authors may move some empirical results in plot to Section 1, to make the readers more easily to jump into the main idea and discovery of the work; 2) the related work part can be extendeded. Specifically, please discuss the difference and relation to the recent work in ICML 2023: On the power of foundation models. As far as I know, the paper also intensively uses the category theory tools for interpreting neural networks especially foundations models. Also, as the paper proposes a new pruning mehtod, the related work part need also discuess peer methods; 3) for experiment part, the authors are required to compare peer pruning methods as the main technical approach presented in this paper is a pruning technique. Also the authors shall more comprehensively discuss the pros and cons of their pruning method in the context of network pruning.

**Questions:**

How sensitive of the proposed IFM method to the hyperparameter \beta? The authors shall discuss it and give some ablation studies if possible.

---

> ### Author Response · Authors · 2023-11-13
> **Response to Reviewer Po5f**
>
> We thank reviewer Po5f for the valuable feedback and suggestions for improvement. We address each point in the comment below.
>
> > **Q1: The authors may move some empirical results in plot to Section 1, to make the readers more easily to jump into the main idea and the discoveries of the work.**
>
> Thanks for the suggestion, we have added an illustration in the Fig.1 in the updated paper to help readers understand the main idea more easily.
>
> > **Q2: The related work part can be extendeded. Specifically, please discuss the difference and relation to the recent work in ICML 2023: On the power of foundation models. As far as I know, the paper also intensively uses the category theory tools for interpreting neural networks especially foundations models. Also, as the paper proposes a new pruning mehtod, the related work part need also discuess peer methods.**
>
> We have extended the related work section in the updated paper, where we discuss and introduce previous works using category theory in machine learning including "On the power of foundation models". Due to the space limitation, we add the discussion about pruning methods in Appendix F in the updated paper.
>
> > **Q3: For experiment part, the authors are required to compare peer pruning methods. Also the authors shall more comprehensively discuss the pros and cons of their pruning method in the context of network pruning.**
>
> We have added the discussion about the difference between our proposed method and other pruning methods as well as the pros and cons in Appendix F in the updated paper. The main difference is that IFM has a different purpose compared with channel pruning methods. The proposed IFM aims at finding the most compact representation without significantly affect the performance while the purpose of channel pruning is better preserving the performance under certain computation budget. The main advantage of IFM is that it does not require access to data while the disadvantage of IFM is that the compression ratio could not be precisely controlled. In the experiment part, we compare our method with another SOTA pruning method INN [1] that also does not require finetuning. Since our proposed method does not require finetuning while most other pruning methods do, it might be unfair to directly compare our results with other pruning methods.
>
> [1] Kirill Solodskikh, Azim Kurbanov, Ruslan Aydarkhanov, Irina Zhelavskaya, Yury Parfenov, Dehua Song, and Stamatios Lefkimmiatis. Integral neural networks. In CVPR, pp. 16113–16122. IEEE, 2023.
>
> > **Q4: How sensitive of the proposed IFM method to the hyperparameter \beta? The authors shall discuss it and give some ablation studies if possible.**
>
> As introduced in the Appendix D, for all the experiments, we apply grid search on $\beta$, from 0.01 to 0.2. The empirical results reported in the experiment part (Fig.3 and Fig.4) reports different results with varing $\beta$. Generally, the \beta set as 0.05 could achieve good balance between the performance and the compression ratio across the experiments we conducted in this paper. We will add a more comprehensive ablation studies in our paper.

---

### Official Review · Reviewer_R8Xc · 2023-10-30

**Soundness:** 2 fair
**Presentation:** 2 fair
**Contribution:** 1 poor
**Rating:** 3
**Confidence:** 4

**Summary:**

The paper proposes an algorithm for merging of functionally equivalent neurons in a neuron network - neurons that do the same job due to having same input weights and same output weights on the same respective input and output connections.  Empirical evaluation is provided showing that sometimes a substantial pruning of the neurons (or merging of features, as the authors call it) can be achieved for a relatively minor drop in accuracy performance.  Authors propose to use the size of the network after the pruning/merge as the means of quantifying network complexity.

**Strengths:**

The method is extremely straight forward - compare the distance between concatenated vector of in-going and out-going weights of two neurons in a layer, and if they are below threshold, declare them the same.

Empirical evaluation seems to show this to be an effective method of pruning neurons in larger networks.

**Weaknesses:**

I don't understand what the point of dragging of the reader through the group theory part of the paper is accomplishing.  It doesn't seem to me like group theory is used to arrive at the proposed feature merging/pruning algorithm.  Permutation of pairs of neurons in a network by swapping their input and output weights (provided they are connected to the same inputs and outputs) is sort of obvious, and in that light, the merging of the features/neurons is quite simple and straight forward.  Defining categories, functors and objects in this setting doesn't seem to give us any extra insight, or produce different tools for how to do the merging.  I find that the theoretical part of the paper has very little to do with the proposed practical aspect, other than perhaps we can name things using group theory terminology.  And if anything, it seems to obfuscate a very straight forward pruning technique that is proposed.

In this race to the state of the art accuracy (which I still think we are all in) it seems that any pruning technique that sacrifices even a fraction of accuracy for some gains in computational/training time costs, is not going to be of practical use.

I can't tell if the proposed feature complexity is meaningful in any way.  Does it correlate with generalisation?

**Questions:**

Aside from my objection to the group theory aspect of the paper, I don't understand what the concept of shape of the features relates to.  Is it the shape of the matrix/tensor representing the features?

Repeating my question from the previous section - is feature complexity related to generalisation?  And what is actually feature complexity?  Is it the total count of neurons in a network after pruning?  How do we know it means something?

The statement "feature complexity corresponds to the most compact representation of a neural network" sound like something related to the minimum description length (MDL) or minimum message length (MML) principles.  Any comment how your work relates to those?

---

> ### Author Response · Authors · 2023-11-13
> **Response to Reviewer R8Xc [1/2]**
>
> We thank reviewer R8Xc for the time and effort devoted to the reviewing process. We address each of the questions in the comments below.
>
> > **Q1: The proposed method is straight forward, what is the point of introducing category theory? The theoretical part of the paper has very little to do with the practical part and doesn't seem to provide any extra insights.**
>
> In the theoretical part of this paper, we define functionally equivalent feature and feature complexity (layer-wise). To facilitate the understanding, we empirically describe them here. Functionally equivalent features mean that, given a feature map of one model, under certain transformations, we could get the output of the model by inferring with another different model that has learned functionally equivalent feature. Take a step further, if we could swap two elements in the feature map while keeping it to be functionally equivalent to the original feature map, it means the two elements could transform to each other, which indicates redundancy in the feature map. Feature complexity is defined as the dimensionality of the most compact representation of the feature without redundancy.
>
> **There are strong connections between the theoretical part and the practical part**:
>
> * With the definition of functionally equivalent features, we theoretically prove that the well-known empirical phenomenon Linear Mode Connectivity (LMC) indicates a special case of two models learning functionally equivalent features.
>
> * We extend the weight matching methods in LMC literature and propose the method IFM to measure feature complexity. Empirical results show that the defined redundancy widely exists in the features learned by neural networks.
>
> In fact, pruning effect is not the main purpose of the proposed method, but a byproduct resulting from the redundancy in features.
>
> **The reasons for introducing category theory**: To formalize the empirical description above in the language of mathematics, we find category theory an appropriate tool.
> * The definition using the language of category theory is concise and elegant in its expression. Two networks learn functionally equivalent features iff there exist a natural transformation between the corresponding functors (formal definition is in Def. 3.1).
> * With our definition of categories, functors and natural transformations in the deep learning context, we hope the category theory perspective of our definition could inspire future works in more theoretical analyses.
> * Despite the use of the terms in category theory, each of them has a corresponding specific meaning in the training or testing of neural networks. Please refer to Table 1 in the updated paper for details.
>
> **To facilitate the understanding of the theoretical part of the paper, revisions have been made.** We add an overview figure (Fig. 1) to empirically illustrate the concepts we introduce in this paper. We also add a table (Table 1) listing the corresponding specific meaning of each used category theory terms in our definition. The revisions in the updated paper are indicated in blue.
>
> > **Q2: It seems that any pruning technique that sacrifices even a fraction of accuracy for some gains in computational/training time costs, is not going to be of practical use.**
>
> While reducing computational cost could be important for scenarios such as edge devices. We would like to clarify that the main purpose of this paper is not neural network pruning. As indicated in the title, we focus on the problem that feature similarity methods sometimes fail to provide sufficient insights (e.g. similar features could also yield different outputs [1,2]). In this paper, we extend the concept of equivalent feature and propose functionally equivalent feature to mitigate the gap between feature similarity and functional similarity (the similarity between outputs)[3]. We further define layer-wise feature complexity corresponding to the dimensionality of the most compact representation of the feature and propose algorithm IFM to measure it. Various insights have been provided (Sections 5.2 and 5.3). Here we list a few insights:
> * Redundancy widely exist in features learned by neural networks.
> * Larger networks have more redundant parameters.
> * Across the layers of a neural network, the feature complexity first increases then decreases.
>
> [1] Frances Ding, Jean-Stanislas Denain, and Jacob Steinhardt. Grounding representation similarity through statistical testing. Advances in Neural Information Processing Systems, 34:1556–1568,2021.
>
> [2] Lucas Hayne, Heejung Jung, Abhijit Suresh, and R McKell Carter. Grounding high dimensional representation similarity by comparing decodability and network performance. 2022.
>
> [3] Max Klabunde, Tobias Schumacher, Markus Strohmaier, and Florian Lemmerich. Similarity of neural network models: A survey of functional and representational measures. arXiv preprint arXiv:2305.06329, 2023.

---

> ### Author Response · Authors · 2023-11-13
> **Response to Reviewer R8Xc [2/2]**
>
> > **Q3: I don't understand what the concept of shape of the features relates to. Is it the shape of the matrix/tensor representing the features?**
>
> Yes, it is the shape of the matrix/tensor representing the features. We have added an description in Sec.2.2 to make it more clear.
>
> > **Q4: What is actually feature complexity? Is it the total count of neurons in a network after pruning? How do we know it means something? Does it correlate with generalisation?**
>
> Ideally speaking, feature complexity is the smallest number of neurons in a network (at a certain layer) that one can achieve without affecting the functionality of the network (loss, accuracy, etc.)
>
> We propose IFM to approximately measure the feature complexity. As we apply grid search on the hyper parameter $\beta$ in Eq. 9, empirical results (Fig. 3(a) and Fig. 4(a) in the updated paper) approximately show the fraction of redundant parameters. These results suggest that redundancy widely exists in features learned by neural networks.
>
> We are not sure if the defined feature complexity is correlated with generalization and we leave it for future works. However, empirical results (Fig. 4(b) in the updated paper) indicate that for two networks of the same structure, the performance drops faster for the network with better performance as more features are merged. We conjecture that networks with higher feature complexity would have better generalization performance.
>
> > **Q5: The statement "feature complexity corresponds to the most compact representation of a neural network" sound like something related to the minimum description length (MDL) or minimum message length (MML) principles. Any comment how your work relates to those?**
>
> The feature complexity we defined in this paper aims to reflect the inherent structure of the neural network, which is the result of training data, network structure and training algorithm. Therefore it does not have direct relationship with MDL or MML principle [1] that guides the model selection or model training. However, the proposed feature complexity could be used as a tool to help the application or interpretation of the minimum message length principle. It could be an interesting topic and we leave it for future works.
>
> [1] Wallace, C. S. (Christopher S.), -2004. (2005). Statistical and inductive inference by minimum message length. New York: Springer. ISBN 9780387237954. OCLC 62889003

---

> > ### Comment · Reviewer_R8Xc · 2023-11-22
> > **Thanks for the response**
> >
> > Thank you for addressing my questions.  The additions you made to the paper definitely improve things, though I am still not quite sure what the implications of this work are and I can't help but feel that, after the grand setup of the category theory, the final algorithm for detecting feature similarity is somewhat trivial - threshold based grid search for Euclidean distance between input and output weights of a neuron.  From the insights, the only one that tells me something new is that across the layers of the network feature complexity first increases then decreases....though then perhaps all it means hat models generally transform input into higher dimensional space before contracting...in which case, perhaps it's not that surprising?
> >
> > And I am a bit puzzled about the statement on the larger feature complexity potentially correlating with generalisation.  I would guess that model that memorises inputs (as in, say, trained to zero training error with randomly shuffled label data) would have massive feature complexity as compared to model that finds common patterns.
> >
> > All in all, I think this is interesting work, that may eventaully lead to substantial insights...but not quite at this stage yet.

---

> > > ### Author Response · Authors · 2023-11-23
> > > **Further Response to Reviewer R8Xc**
> > >
> > > We thank reviewer R8Xc for the response to our rebuttal. As Reviewer R8Xc has noted in the comment, our proposed algorithm is indeed simple and straight forward. However, we believe simple algorithm does not necessarily mean trivial. **The proposed algorithm is deeply connected to the category theory framework we proposed.** On the one hand, our algorithm is rooted in the proposed mathmatical framework. Motivated by this proposed framework, we derive and propose the simple and effective algorithm. On the other hand, **without a specific design, we use an extremely simple algorithm to empirically prove that the defined redundancy in learned features widely exists.** With the mathmatical framework and the defined feature complexity, we offer a new perspective for better understanding the features learned by neural networks.
> > >
> > >
> > > When comparing the proposed method to pruning methods, in the newly added Table 3 in Appendix F, we demonstrate that **, to our best knowledge, our proposed method is the first that does not require any access to data (including finetuning and intervening the training process).**
> > >
> > > Comparing our work to previous works using category theory, **we derived the specific algorithm that could be of practical use on modern neural networks while most previous works solely provide their construction in category theory without specific algorithms**, e.g. [1] defines categroid to study conditional independence, [2] shows that, under certain condition, gradient decent is a monoidal functor from a category of parameterised functions to a category of update rules, [3] characterizes the invariances of learning algorithms using category theory and illustrate the framework on linear regression.
> > >
> > > Regarding the topic of generalization, as we have mentioned in the rebuttal, we have not explored this topic and we are not sure if the defined feature complexity is correlated with the generalization performance of the neural network. It is indeed possible that neural networks with higher feature complexity memorizes the data while neural networks with lower feature compelxity learns common patterns. **In fact, our proposed feature complexity could be a helpful tool for future works studying generalization in determining whether the neural network memorizes data or learns common patterns.**
> > >
> > > [1] Sridhar Mahadevan. Categoroids: Universal conditional independence. arXiv preprint arXiv:2208.11077, 2022a.
> > >
> > > [2] Brendan Fong, David Spivak, and Remy Tuy ´ eras. Backprop as functor: A compositional perspective on supervised learning. In 2019 34th Annual ACM/IEEE Symposium on Logic in Computer Science (LICS), pp. 1–13. IEEE, 2019.
> > >
> > > [3] Kenneth D Harris. Characterizing the invariances of learning algorithms using category theory. arXiv preprint arXiv:1905.02072, 2019.

---

### Official Review · Reviewer_QeEM · 2023-10-31

**Soundness:** 3 good
**Presentation:** 2 fair
**Contribution:** 2 fair
**Rating:** 6
**Confidence:** 4

**Summary:**

The paper attempts to tackle a pertinent issue in neural networks, focusing on understanding and measuring the similarity between features learned by different neural network initializations. The concepts presented seem promising, especially in defining what are termed as functionally equivalent features and the subsequent proposal of an algorithm, Iterative Feature Merging (IFM). However, there are concerns and areas of improvement that need addressing.

**Strengths:**

* The paper addresses a significant issue in neural networks—understanding the behavior and feature representation across different initializations.

* The introduction of the term "functionally equivalent features" offers a fresh perspective on understanding neural networks.

* The Iterative Feature Merging (IFM) seems promising as an approach to quantify feature complexity.

**Weaknesses:**

* The term "functionally equivalent features" is central to the paper. It would be beneficial to the reader if a simple illustrative example or intuitive explanation accompanied its introduction.

* While the abstract and conclusion provide a concise overview, it remains unclear how rigorous the definitions and proofs, especially related to category theory, are in the main content of the paper. The paper would benefit from a detailed walkthrough of the mathematical formulations and proofs to ensure the robustness of the claims made.

* Given the recent interest in understanding neural network behavior and their feature representations, it is vital to contextualize this work concerning existing literature. How does this work differ or extend previous work on the topic? A comprehensive comparison is essential.

**Questions:**

* The conclusion acknowledges a limitation in testing only for image classification tasks. Why was the scope of experiments limited to image classification tasks, and how would the approach perform on other tasks? It would strengthen the paper to include experiments from a wider range of tasks or at least provide a rationale for why image classification was chosen as the primary focus.

* Can the authors provide a more intuitive or illustrative example of "functionally equivalent features" to aid understanding?

* The Iterative Feature Merging (IFM) algorithm is a central piece of this work. How does the Iterative Feature Merging (IFM) algorithm work in detail, and what makes it efficient? A clear and detailed algorithmic procedure, possibly with pseudocode, should be provided for a comprehensive understanding.

---

> ### Author Response · Authors · 2023-11-13
> **Response to Reviewer QeEM [1/2]**
>
> We thank reviewer QeEM for the valuable feedback and suggestions for improvement. We address each point in the comment below.
>
> > **Q1: The term "functionally equivalent features" is central to the paper. It would be beneficial to the reader if a simple illustrative example or intuitive explanation accompanied its introduction.**
>
> Thanks for the suggestion. We have added an intuitive illustration (Fig.1) with intuitive explanation of "functionally equivalent feature" and "feature complexity" in the updated paper.
>
> > **Q2: The paper would benefit from a detailed walkthrough of the mathematical formulations and proofs to ensure the robustness of the claims made.**
>
> Thanks for the suggestion. In our main submission, we provided detailed proofs in Appendix C. We also add a more detailed walkthrough of our formulation with category theory in Appendix C.3 of the updated paper.
>
> > **Q3: It is vital to contextualize this work concerning existing literature. How does this work differ or extend previous work on the topic? A comprehensive comparison is essential.**
>
> There are two perspectives in the literature investigating similarity between neural networks, feature similarity [1,2,3,4] and functional similarity [5,6,7]. Feature similarity (representational similarity) focuses on the similarity between intermediate features while functional similarity focuses on the similarity between outputs [8]. However, evidences have shown that similar features could yield different output [9].
>
> In this paper, we unify the two perspectives and propose functionally equivalent feature. Based on functionally equivalent feature, we further define feature complexity and propose an algorithm IFM to measure the feature complexity. The proposed algorithm is an extension of weight matching methods [10] in linear mode connectivity literature from between two neural networks to within one neural network.
>
> In response to the suggestion, a more detailed introduction and comparison have been added in Sec. 5 in the updated paper.
>
> [1] Maithra Raghu, Justin Gilmer, Jason Yosinski, and Jascha Sohl-Dickstein. SVCCA: singular vector canonical correlation analysis for deep learning dynamics and interpretability. In NIPS, pp. 6076–6085, 2017
>
> [2] Alex H. Williams, Erin Kunz, Simon Kornblith, and Scott W. Linderman. Generalized shape metrics on neural representations. In NeurIPS, pp. 4738–4750, 2021
>
> [3] Shuai Tang, Wesley J. Maddox, Charlie Dickens, Tom Diethe, and Andreas C. Damianou. Similarity of neural networks with gradients. CoRR, abs/2003.11498, 2020.
>
> [4] Serguei Barannikov, Ilya Trofimov, Nikita Balabin, and Evgeny Burnaev. Representation topology divergence: A method for comparing neural network representations. In ICML, volume 162 pp. 1607–1626. PMLR, 2022.
>
> [5] Omid Madani, David Pennock, and Gary Flake. Co-validation: Using model disagreement on unlabeled data to validate classification algorithms. Advances in neural information processing systems, 17, 2004.
>
> [6] Yamini Bansal, Preetum Nakkiran, and Boaz Barak. Revisiting model stitching to compare neural representations. Advances in neural information processing systems, 34:225–236, 2021.
>
> [7] Srinadh Bhojanapalli, Kimberly Wilber, Andreas Veit, Ankit Singh Rawat, Seungyeon Kim, Aditya Menon, and Sanjiv Kumar. On the reproducibility of neural network predictions. arXiv preprint arXiv:2102.03349, 2021.
>
> [8] Max Klabunde, Tobias Schumacher, Markus Strohmaier, and Florian Lemmerich. Similarity of neural network models: A survey of functional and representational measures. arXiv preprint arXiv:2305.06329, 2023.
>
> [9] Frances Ding, Jean-Stanislas Denain, and Jacob Steinhardt. Grounding representation similarity through statistical testing. Advances in Neural Information Processing Systems, 34:1556-1568,2021.
>
> [10] Samuel Ainsworth, Jonathan Hayase, and Siddhartha Srinivasa. Git re-basin: Merging models modulo permutation symmetries. In ICLR, 2023.
>
> > **Q4: Why was the scope of experiments limited to image classification tasks, and how would the approach perform on other tasks?**
>
> We conduct experiments on image classification tasks following previous feature similarity literature and pruning literature. In those literature, image classification task is the most widely used and investigated. To align with previous works, our experiments are conducted on most widely used network structures (VGG and ResNet) and benchmarks (CIFAR10 and ImageNet-1K).
>
> Our definition does not have any assumption about the neural network structure and the proposed method does not require access to data. While our focus on image classification tasks aligns with the prevalent literature, the task-agnostic nature of our method suggests its potential applicability to various tasks. Considering the limited time and space, we leave the approach on other tasks for future works.

---

> ### Author Response · Authors · 2023-11-13
> **Response to Reviewer QeEM [2/2]**
>
> > **Q5: Can the authors provide a more intuitive or illustrative example of "functionally equivalent features" to aid understanding?**
>
> Yes, an illustrative figure has been added in the updated paper (Fig. 1). More intuitively, if two models learn functionally equivalent features, it means that, under certain transformations, we can replace any layer of one model with the corresponding layer of the other model and get the same output as the original model.
>
> > **Q6: How does the Iterative Feature Merging (IFM) algorithm work in detail, and what makes it efficient? A clear and detailed algorithmic procedure, possibly with pseudocode, should be provided for a more comprehensive understanding.**
>
> In the main submission, we provided detailed algorithm in Appendix. A. To facilitate the understanding of the proposed method, we have added a flowchart in Fig. 1 in the updated paper which is indicated in blue. In general, the proposed method is efficient because it does not require the access to data and only matches the weights of neural networks.

---

> > ### Comment · Reviewer_QeEM · 2023-11-23
> >
> > Thanks for the response. The authors have addressed most of my concerns and I'll raise my evaluation to '6: marginally above the acceptance threshold'.

---

### Official Review · Reviewer_oxNv · 2023-10-31

**Soundness:** 3 good
**Presentation:** 3 good
**Contribution:** 3 good
**Rating:** 6
**Confidence:** 4

**Summary:**

This paper proposes the concept of neural network feature complexity which is (claimed by the authors) a more formal and comprehensive description of the behavior of the trained neural networks. This idea, along with the technical part including theory, approach and empirical results, to my best knowledge, are basically new in literature. Specifically, the authors develop a quantitative metric to enable the reduction of neural network parameters based on the proposed concept of feature equivalence. In fact, the layer-wise feature complexity is also well supported by a number of empirical studies which also well align with the intuition: e.g. higher feature complexity with the same size of networks can achieve better performance; a larger network tends to learn redundant feaures; equivalents features may correspond to certain low-level semantics. It also well establishes the connection to linear mode connectivity (LMC).

One other interesting aspect of the paper is introducing the powerful tool of category theory to establish the (clear) theoretical foundation of the work: represent the network structure as a category and a certain neural network as a functor that maps this structure to specific parameters. The authors also give a clear discussion to differentiate their work for using the category theory. This perspective along with its technical derivation and results are new to my best knowledge and well fit with the proposed paradigm for understanding the feteature complexity.

**Strengths:**

The authors have well stated the contributions in their paper and my main concerns are mostly from writing and presentation. I think the authors are very familiar to the field of the paper and their presented theoretical results and methods are new, well-motivated and well verified by the experiments (espeically from the pruning perspective).

**Weaknesses:**

Yet the paper need to be more self-contained when introducing the ideas and background There are a few specific suggestions:
1) in the abstract, the authors are suggested to emphasize the feature complexity is layer-wise to make it more clear as this is a new concept to my knowledge; they may use the term layer-wise ASAP in the abstract.
2) it is vague to use the saying: merge of features in the first paragraph. Its exact and clear meaning need be better explained for self-containess. I think it refers to merge the weight to merge the feature;
3) I am a bit confused with Eq. 4, how \tau_z^{l+1} is imposed on Z^{l+1}? Is it a dot operation or we shall write it as a function of \tau_z^{l+1}?
4) it would also be good to clarify what are the Functionally Equivalent Features in Definitions 3.1, I think they are layer-wise f^{l}(theta_a^{l}) vs. f^{l}(theta_b^{l})?
5) it a bit abuses the notations? does Z() also equals to f()? Please clarify it.
6) in the introduction part, the authors may mention the partial order idea to make the complexity concept more tangible to readers, rather than until give the specific definition in Section 3.

some minor typos:
In another words->In another word
it simply apply no transformation->it simply applies no transformation
in definition 3.1 Functionally Equivalent Feature->Functionally Equivalent Features
at l-th layer of network->at the l-th layer of network
in Theorem 3.3 to a itself->to itself

Finally I suggest the authors consider to put an overview of their work in the introduction part to improve the readability.

**Questions:**

How the proposed pruning comapre with other (SOTA) pruning method? Additional experiments will further enhance the paper.

---

> ### Author Response · Authors · 2023-11-13
> **Response to Reviewer oxNv**
>
> We thank reviewer QeEM for the valuable feedback and suggestions for improvement. According to the suggestions, we have added an overview figure in the updated paper and fixed the mentioned minor typos. We address each point in the comment below.
>
> > **Q1: In the abstract, the authors are suggested to emphasize the feature complexity is layer-wise to make it more clear as this is a new concept to my knowledge; they may use the term layer-wise ASAP in the abstract.**
>
> Thanks for the suggestion, we have revised the introduction (Sec.1) and emphasized that the feature complexity is layer-wise.
>
> > **Q2: it is vague to use the saying: merge of features in the first paragraph. Its exact and clear meaning need be better explained for self-containess.**
>
> We have revised the introduction section and make it more clear that we merge the weight to merge the feature.
>
> > **Q3: I am a bit confused with Eq. 4, how $\tau_z^{l+1}$ is imposed on $Z^{l+1}$? Is it a dot operation or we shall write it as a function of $\tau_z^{l+1}$?**
>
> Sorry for the confusion. In Eq. 4, $\tau_z^{l+1}$ is a function. We followed the category theory literature and omitted parentheses for simplicity. We have added parentheses to avoid confusion.
>
> > **Q4: It would also be good to clarify what are the Functionally Equivalent Features in Definitions 3.1, I think they are layer-wise $f^{l}(\theta_a^{l})$ vs. $f^{l}(\theta_b^{l})$?**
>
> Thanks for the suggestion. We have added an illustration figure in the updated paper (Fig. 1). However, since the input of $f^{l}$ is determined by the previous $l-1$ layers, the Functionally Equivalent Features in Def. 3.1 is not defined layer-wise but "model-wise".
>
> > **Q5: Is it a bit abuses the notations? does Z() also equals to f()? Please clarify it.**
>
>
> In our notation, $Z^{l}$ represent the composition of the first $l$ layers of the network. We use $Z^{l}$ instead of $f^{l}(f^{l-1}(\cdots f^{1}()))$ for simplicity.
>
> > **Q6: In the introduction part, the authors may mention the partial order idea to make the complexity concept more tangible to readers, rather than until give the specific definition in Section 3.**
>
> Thanks for the suggestion. We have added an illustration and intuitive description of the feature complexity in Fig. 1 in the updated paper to make the complexity concept more tangible.
>
> > **Q7: How the proposed pruning comapre with other (SOTA) pruning method? Additional experiments will further enhance the paper.**
>
> We have added an detailed discussion of differences between the proposed IFM and other pruning methods in Appendix F in the updated paper. The comparision with another SOTA pruning method INN [1] that also does not require finetuning was provided in the experiment section. Since our proposed method does not require finetuning while most other pruning methods do, it might be unfair to directly compare our results with other pruning methods.
>
> [1] Kirill Solodskikh, Azim Kurbanov, Ruslan Aydarkhanov, Irina Zhelavskaya, Yury Parfenov, Dehua Song, and Stamatios Lefkimmiatis. Integral neural networks. In CVPR, pp. 16113–16122. IEEE, 2023.

---

> > ### Author Response · Authors · 2023-11-23
> > **Looking forward to further feedback**
> >
> > Dear Reviewer oxNv,
> >
> > We would like to express our heartfelt gratitude for your detailed and constructive comments. According to the comments, we have refined our paper, provided an overview figure (Fig. 1) to facilitate understanding of our proposed framework and have added a detailed discussion of differences between our proposed method and other pruning methods in Appendix F. As the deadline of discussion period is approaching, we would appreciate your feedback about whether our rebuttal has addressed your concerns. We would be more than grateful if the rating could be reconsidered after our rebuttal.
> >
> > Best regards,
> >
> > The Authors

---

### Author Response · Authors · 2023-11-13
**General Response to All Reviewers**

We appreciate the time and effort reviewers devoted to the reviewing process. Overall, reviewers consider the concepts we proposed as new (oxNv, QeEM, Po5f) and our proposed method as effective (oxNv, QeEM, R8Xc, Po5f). We have individually responded to each reviewer’s comments and questions, addressing them in our rebuttal. In response to the comments, we have made revisions in the updated paper. These revisions are indicated in blue.

To facilitate a clearer understanding of our proposed concepts and methods:
* **An overview and more intuitive description of our proposed concepts and methods have been added (Fig.1).**
* **A table listing the terms in category theory and their corresponding specific concepts in our definition has been included (Table 1).**
* **A more detailed walkthrough of our formulation using category theory has been added in Appendix.B.3.**

To better introduce the background, contextualize our work and compare with previous works:
* **The related works section has been extended to better contextualize our work. We also provide an introduction and comparison with previous work using category theory in machine learning.**
* **A discussion about our proposed method IFM and other pruning methods has been added in Appendix F.**

Other changes have been made to accommodate the revisions and fix minor typos. We are happy to respond to further questions and comments and will do our best to address all concerns. If there are additional concerns, please let us know, and we will promptly update our response.

---

### Author Response · Authors · 2023-11-19
**Inquiry for post-rebuttal comments**

Dear reviewers,

We appreciate your valuable comments on our paper. Since the discussion period is approaching its end, we would be more than happy to hear from you about whether our rebuttal has addressed your concerns. If you have any further questions and concerns, feel free to post any comments. We are happy to respond further questions and comments and we will try our best to address all the concerns.

---

### Author Response · Authors · 2023-11-20
**Looking forward to further discussion**

Dear reviewers,

We would like to express our sincere gratitude again  for your valuable comments and thoughtful suggestions. Throughout the rebuttal phase, we tried our best to address concerns, augment experiments to fortify the paper (comprising approximately 3 pages of new content with 2 new tables and 1 new figure), and refine details in alignment with your constructive feedback. Since the discussion time window is very tight and is approaching its end, we truly hope that our responses have met your expectations and assuaged any concerns. We genuinely do not want to miss the opportunity to engage in further discussions with you, which we hope could contribute to a more comprehensive evaluation of our work. Should any lingering questions persist, we are more than willing to offer any necessary clarifications.

With heartfelt gratitude and warmest regards,

The Authors

---

### Comment · Area_Chair_RhT4 · 2023-11-21
**Authors-Reviewers Discussion Phase**

Dear Reviewers,

I encourage you to use this opportunity to engage with the authors if you have any remaining concerns regarding this paper. Thank you for your dedicated service.

Sincerely,
AC

---

### Meta-Review · Area_Chair_RhT4 · 2023-12-08

**Metareview:**

The paper introduces the concept of "neural network feature complexity" to evaluate trained neural networks. It proposes a metric for feature equivalence, supporting the idea that networks with higher complexity can be more efficient. Using category theory for theoretical backing, the paper presents the Iterative Feature Merging (IFM) algorithm, which merges functionally equivalent neurons for network pruning with minimal accuracy loss. This approach provides insights into feature similarity and equivalence, offering a practical method for measuring and reducing network complexity.

**Justification For Why Not Higher Score:**

In their response, the authors address concerns regarding the significance of using category theory to evaluate deep neural networks and its connection to their algorithm. They clarify that within the framework of category theory, they derive definitions for functionally equivalent features and feature complexity (layer-wise). This theoretical foundation leads to the proof that the well-known empirical phenomenon of Linear Mode Connectivity (LMC) represents a special case where two models learn functionally equivalent features. Building upon the weight matching methods in LMC literature, the authors propose the Iterative Feature Merging (IFM) method to measure feature complexity. Their empirical results demonstrate that the defined redundancy is prevalent in features learned by neural networks. While the resulting algorithm may appear simple, it is deeply rooted in and well motivated by the authors' comprehensive evaluation of neural networks through a category theory perspective.

**Justification For Why Not Lower Score:**

Based on the review comments, this paper merits acceptance due to its innovative approach in introducing the concept of "neural network feature complexity," a novel method for analyzing trained neural networks. The integration of category theory for a robust theoretical foundation is particularly notable. Furthermore, the paper's development of the Iterative Feature Merging (IFM) algorithm represents a significant advancement, demonstrating how functionally equivalent neurons can be merged for effective network pruning with minimal loss in accuracy. These contributions offer valuable insights into feature similarity and equivalence in neural networks, presenting a unique method for reducing network complexity.

---

### Decision · Program_Chairs · 2024-01-16

Accept (poster)